# Spectroscopy of a mesospheric ghost reveals iron emissions

María Passas-Varo [1], Oscar Van der Velde [2], Francisco J. Gordillo-Vázquez[1], Juan Carlos Gómez-Martín[1], Justo Sánchez[1], Francisco J. Pérez-Invernón[1], Rubén Sánchez-Ramírez[1], Maya García-Comas[1] & Joan Montanyà[2]

Mesospheric Green emissions from excited Oxygen in Sprite Tops (ghosts) are infrequent and faint greenish transient luminous events that remain for hundreds of milliseconds on top of certain energetic sprites. The main hypothesis to explain this glow persistence is the long lifetime of excited atomic oxygen at 557.73 nm, a well-known emission line in aurora and airglow. However, due to the lack of spectroscopic campaigns to analyse such events to date, the species involved in the process can not yet be identified. Here we report observational results showing the temporal evolution of a ghost spectrum between 500 nm and 600 nm. Besides weak -but certain- traces of excited atomic oxygen, our results show four main contributors related to the slow decay of the glow: atomic iron and nickel, molecular nitrogen and ionic molecular oxygen. Additionally, we are able to identify traces of atomic sodium, and ionic silicon, these observations being consistent with previous direct measurements of density profiles of meteoric metals in the mesosphere and lower thermosphere. This finding calls for an upgrade of current air plasma kinetic understanding under the influence of transient luminous events.

Transient luminous events (TLEs) are huge and brief optical and electrical phenomena occurring over thunderstorms, in the upper regions of the Earth's atmosphere, after high energetic lightning activity. They are also known as upper atmospheric lightning, and the most common TLEs include sprites[1], halos[2], elves[3], blue jets[4], and gigantic jets[5].

TLEs were serendipitously observed for the first time in 1989[6] and, since then, sky photographers all around the globe spend the nights chasing and sharing these breathtaking flashes of light. It was May 2019 when, while documenting a sprite storm over Oklahoma, a citizen scientist realised that a greenish glow appeared on top of certain energetic sprites and lasted for several milliseconds[7]. This new phenomenon was named then from the acronym of Green emissions from excited Oxygen in Sprite Tops (ghost) because, up to now, the main hypothesis to explain its greenish colour is the excited atomic oxygen at 557.73 nm[8,9], a well-known emission line in aurora and airglow[10]. Since the uploading of this finding to YouTube[11], the TLEs community started to pay attention to these rare greenish emissions[8].

In June 2019, we initiated a systematic spectroscopic campaign aiming to understand the chemistry and dynamics related to ghosts. The objective was to analyse the temporal evolution of the spectral emissions occurring on top of sprites, in a wavelength range from 500 to 600 nm.

In this paper, we show the spectrum of a mesospheric ghost and its temporal evolution. Although the results confirm weak traces of excited atomic oxygen (O I) at 557.57 nm, we are able to identify four main species associated with the gradual fading of the ghost brightness: atomic iron (Fe I), nickel (Ni I) and nitrogen (N I), molecular nitrogen ($N_2$) and ionic molecular oxygen ($O_2^+$). Besides, our results show traces of atomic sodium (Na I) and ionic silicon (Si II). This finding would require an upgrade of current air plasma kinetic models to predict the optical emissions of TLEs[12–15] because, so far, there is an absence of metallic species in their implementation.

[1]Instituto de Astrofísica de Andalucía (IAA), CSIC, Glorieta de la Astronomía sn, 18008 Granada, Spain. [2]Department of Electrical Engineering, Universitat Politécnica de Catalunya, Colom 1, 08222 Terrassa, Spain. ✉e-mail: passasv@iaa.csic.es

## Results

### Imaging and lightning activity

The spectra we report here were observed with an upgraded version of the GRAnada Sprite Spectrograph and Polarimeter (GRASSP)[16–19]. This intensified slit spectrograph is located at 41.677°N, 1.840°E, elevation 276 m above the sea level, in Castellgalí, Barcelona province, Spain.

Since June 2019 we have recorded the images and spectra of 42 sprites with the spectrograph slit projection overlapping their top diffuse region, where mesospheric ghosts are likely to appear. From these 42 events, only one showed a spectrum brightness with a significant signal to noise ratio. The event we refer to occurred on 21 September 2019 on top of a jellyfish sprite that emerged from a thunderstorm cell developed over the Mediterranean Sea, at approximately 370 km distance and 84.1 degrees azimuth from the observation site, as Fig. 1 shows. The LIghtning detection NETwork (LINET)[20] detected the 178.2 kA peak current cloud-to-ground parent stroke (41.9324°N,6.2828°E) at 19:44:30.514 UTC. Table 1 shows LINET lightning activity on 21 September 2019 from 19:44:29.765 UTC to 19:44:32.123 UTC within the area of interest.

A video camera with a 720 nm infrared-pass filter was used in time-lapse mode at 1 second intervals to capture high-resolution images of sprites and the hydroxyl (OH*) airglow background. In these images, we identified a thin cirrus cloud band across part of the sprite, which did not affect the region where the spectra were taken (panel a of Fig. 2). The banded structure visible in the upper part of both panels of Fig. 2 shows a slight curvature consistent with concentric gravity waves

propagating away from the convective core of the thunderstorm system that produced the sprites[21–23].

Panel b of Fig. 2 shows the composite image of the reported event. We estimated the dimensions of the sprite as, approximately, 47 km height and 60 km width and the altitude of the slit projection around $82.8 \pm 0.4$ km, according to the star fitting method[24]. Unfortunately, the infrared long pass filter at 720 nm and the 1 s exposure time of the imaging camera prevented us from detecting the ghost in the frames before and after the sprite. However, the intensified spectrograph camera showed an increase of the brightness in the spectral range between 500 and 600 nm before, during and after the sprite, implying a previous slow enhancement and further decay of green emissions from the top of the sprite.

### Line identification

We analysed 45 consecutive spectra of the reported event, from − 480 ms to 1280 ms, being t = 0 ms the time coincident with the sprite. We detected a global enhancement of the spectral brightness starting approximately 200 ms before the occurrence of the sprite and continuing for 480 ms thereafter. We observed two distinct stages of this brightness enhancement. In Table 2 (set A), the identified emission lines (see methods subsection line identification) enhanced their brightness level during the initial stage, from − 200 ms to 120 ms. Meanwhile, in Table 3 (set B), the emission lines showed an enhancement in their brightness level from 120 to 480 ms, corresponding to the secondary stage. We identified the O I (557.73 nm), Fe I (522.06 nm, 529.07, 535.63 nm, 538.22 nm, 563.96 nm, 580.45 nm, 583.46 nm and

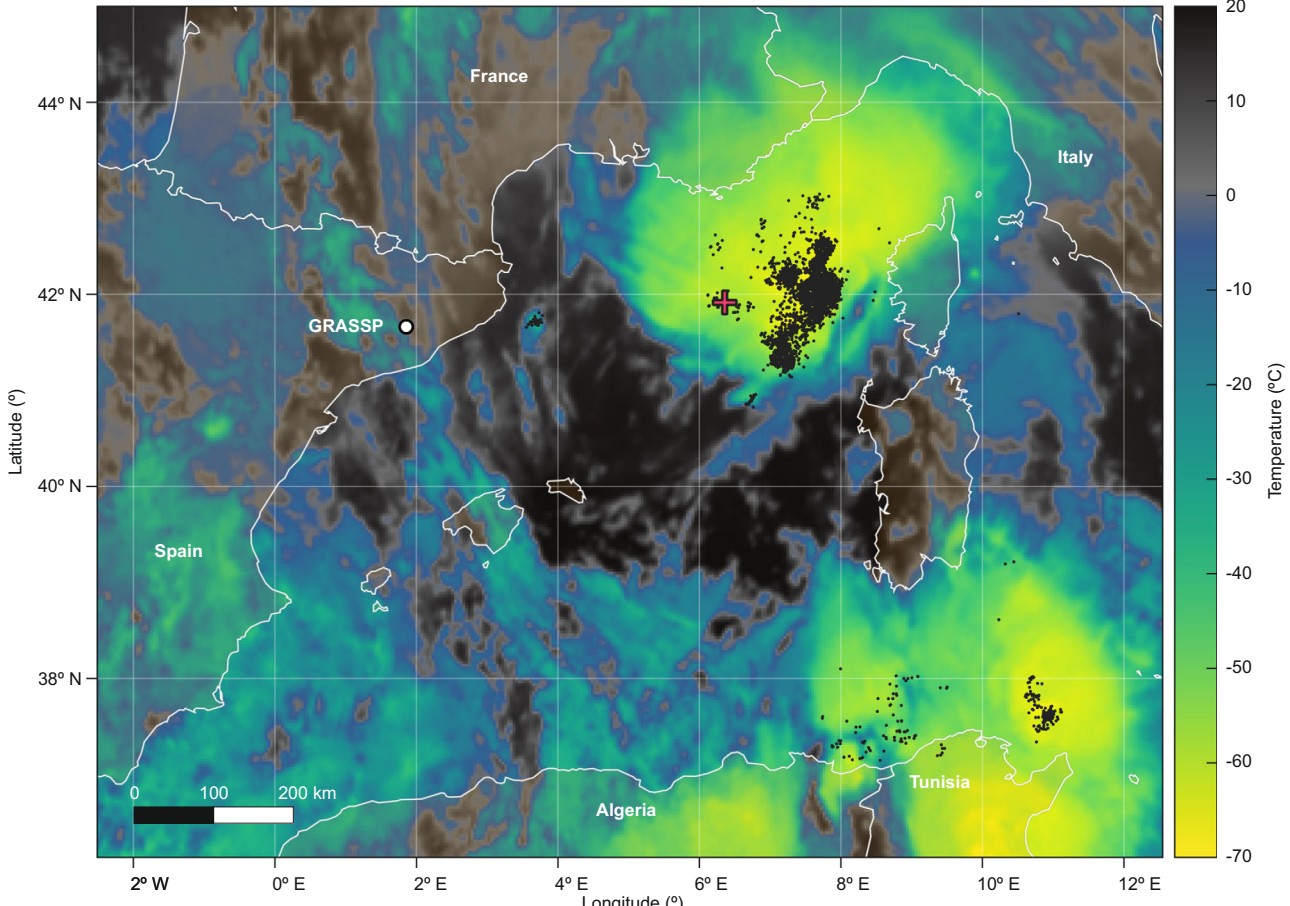

**Fig. 1 | Meteosat Second Generation infrared image (1 km resolution), showing cloud top temperatures on 21 September 2019 19:45 UTC.** Locations of lightning activity from LIghtning detection NETwork (LINET) (black dots), observation site (white circle) and the reported mesospheric ghost parent stroke (red cross) are superimposed.

**Table 1 | Lightning activity**

| Time (UTC) | Latitude (°N) | Longitude (°E) | Type | Peak current (kA) |
|---|---|---|---|---|
| 19:44:29.765 | 42.0432 | 7.6044 | CG | −4.2 |
| 19:44:30.149 | 41.6559 | 7.0774 | CG | −4.8 |
| 19:44:30.150 | 41.6674 | 7.0486 | CG | 10.5 |
| 19:44:30.372 | 41.5852 | 6.9506 | CG | 15.1 |
| 19:44:30.403 | 41.7864 | 6.9443 | CG | 13.4 |
| 19:44:30.403 | 41.7897 | 6.9567 | CG | 13.5 |
| 19:44:30.514 | 41.9324 | 6.2828 | CG | 178.2 |
| 19:44:30.548 | 41.8934 | 6.6062 | CG | 7.1 |
| 19:44:30.550 | 41.8800 | 6.6150 | CG | 18.5 |
| 19:44:30.608 | 41.9181 | 6.6422 | CG | 4.7 |
| 19:44:30.629 | 41.8884 | 6.6076 | CG | 5.5 |
| 19:44:30.629 | 41.8879 | 6.6300 | CG | 5.5 |
| 19:44:30.898 | 42.1914 | 6.6493 | CG | −4 |
| 19:44:31.029 | 41.9795 | 6.1373 | CG | −6.5 |
| 19:44:31.029 | 41.9810 | 6.1718 | CG | −7.4 |
| 19:44:31.160 | 42.3882 | 7.6264 | IC | 3.7 |
| 19:44:32.123 | 42.0152 | 7.5206 | CG | 7.3 |

Lightning activity from 19:44:29.765 UTC to 19:44:32.123 UTC on 21 September 2019. Data provided by the LIghtning detection NETwork (LINET). CG and IC refer to cloud-to-ground and intracloud lightning, respectively.

586.72 nm) forbidden atomic emission lines, and the $N_2$ Vegard Kaplan (VK) molecular emission band ($v' = 0$, $v'' = 15$; 560.80 nm). Their radiative lifetimes are longer than 0.7 s[25,26]. We also identified the $O_2^+$, Si II and Fe II ionic, and N I, Na I, Fe I and Ni I atomic emission lines, their radiative lifetimes being very short[25,26].

We calculated the average of the consecutive reduced spectra of the reported event from −200 ms to 480 ms to detect the species that contribute the most to the spectral brightness enhancement (panel a of Fig. 3). We highlighted four emission lines of this average with brightness level higher than $5\sigma$ (Fe I at 528.04 nm, Fe I at 583.46 nm, $O_2^+$ at 590.00 nm and $O_2^+$ at 599.10 nm) and the O I emission line at 557.73 nm with brightness level higher than $3\sigma$. Panels b and c of Fig. 3 show a zoom of the spectra from 0 ms to 80 ms.

### Line flux

We calculated the line flux (see methods subsection line flux) evolution of the identified species in Tables 2 and 3. Right axis of Fig. 4 shows the temporal evolution (from − 480 to 1280 ms) of the normalised average of the line fluxes of the identified species in combined set A and set B (black solid line). Previous imaging research on mesospheric ghosts reports green emissions to follow similar behaviour[8]: a gradual enhancement reaching its maximum coincident with the sprite occurrence, followed by a further and abrupt decay and a final increase leading to subsequent stabilisation.

If we focus on the identified species at the initial stage (set A), we observe an increase in their average line flux (dotted black line, right axis of Fig. 4) from 200 ms before the event occurrence, reaching its maximum level, and then abruptly decaying until 120 ms afterwards, when it stabilises.

On the other hand, we observe a gradual increase in the average line flux of the detected species during the secondary stage (set B) from 200 ms before the event occurrence to continue up to 320 ms afterwards, reaching its maximum level. Subsequently, it smoothly decays until 480 ms, at which point it stabilises (dashed black line, right axis of Fig. 4).

Left axis of Fig. 4 shows the temporal evolution of the Fe I ($x^3P_2^0 - b^3D_3$) (528.04 nm), O I ($^1S_0 - ^1D_2$) (557.73 nm), $N_2$ ($A^3\Sigma_u^+ - X^1\Sigma_g^+$) ($v' = 0$, $v'' = 15$) (560.80 nm), and Fe I ($a^5P_3$ - $a^5D_3$) (583.46 nm) line flux ratios.

The emission line that contributes the most to the average observed line flux at the initial stage is the forbidden Fe I ($a^5P_3$ - $a^5D_3$) (583.46 nm), reaching a maximum level of approximately 2.7% of the average line flux at 0 ms; the emission line that makes the most significant contribution to the average observed line flux at the secondary stage is Fe I ($x^3P_2^0 - b^3D_3$) (528.04 nm), reaching a maximum level of approximately 4.5% of the average line flux at 160 ms.

## Discussion

Early spectroscopy of TLEs indicated that the only detected emissions in sprites were due to the $N_2$ first positive system[27,28]. Mende et al. (1995)[27] used a vertical slit to study the optical emissions along a sprite, but the spectrum they analysed (from 450 to 800 nm) was not spatially resolved. Therefore, these optical features were integrated in the spatial dimension, and the greenish and bluish emissions - if any - might be negligible compared to the reddish. On the other hand, Hampton et al. (1996)[28] used a horizontal slit to analyse the optical emissions of TLEs from 550 nm to 850 nm, aiming to the main body of a sprite, this region being typically red.

Furthermore, spectroscopy of jets revealed that the blue emissions from 320 to 460 nm were due to the $N_2$ second positive system, and negligible emissions of the $N_2^+$ first negative system were also detected[29].

Spectroscopy of halos between 550 and 850 nm revealed the strong reddish emissions of the $N_2$ first positive system also present in sprites spectrum, but also weak (and ignored) greenish emissions[2]. The authors focused only on the analysis of the $N_2$ emissions, and the spectral resolution of 6 nm prevented them from identifying any isolated emission lines.

Thereafter, TLEs spectroscopic research[2,12–15,19,29–37], targeted the red and blue emissions of $N_2$, in line with the colour of the TLEs images so far. Meanwhile, the green spectral region of TLEs remained unobserved.

After the broadcast of the first mesospheric ghosts video images[11], we started a systematic spectroscopic campaign to analyse the temporal evolution of the spectrum of these phenomena, in the spectral range between 500 and 600 nm, with a spectral resolution higher than 0.31 nm. We recorded the image and the spectral evolution of a sprite, with the spectrograph slit aiming to its top diffuse region, where mesospheric ghosts are probable to develop. We observed some banded structure in the airglow background in the 10 s before and after the sprite time, which we interpret as likely caused by modulation of the hydroxyl (OH*) airglow layer by gravity waves propagating upward from the parent thunderstorm. Indeed, gravity waves are known to modulate atmospheric density at mesospheric altitudes, and to influence the structure of the optical brightness, not only of the airglow layer, but also of sprites[38,39] and elves[40]. Moreover, there are imaging reports of rippling wave-like patterns of green emissions close to the lower ionosphere, that are directly related to gravity waves[7].

In the spectral analysis, we identified nine forbidden atomic emission lines (O I at 557.73 nm and Fe I at 522.06 nm, 529.07, 535.63 nm, 538.22 nm, 563.96 nm, 580.45 nm, 583.46 nm and 586.72 nm) and the molecular emission band $N_2$ (VK) at 560.80 nm, their radiative lifetimes being longer than 0.7 s. We also identified the $O_2^+$, Si II and Fe II ionic, and N I, Na I, Fe I and Ni I atomic emission lines, their radiative lifetimes being very short[25,26].

The $N_2$ VK ($v' = 0$, $v'' = 15$) (560.80 nm) optical transition observed in the spectra is driven from the direct electron-impact excitation of $N_2$ ($X^1\Sigma_g^+$). In fact, previous modelling of the vibrational kinetics of air plasmas produced by the presence of sprites in the mesosphere of the Earth at 78 km predicts the population of the vibrational levels of $N_2$ ($A^3\Sigma_u^+$) to be dominant over the population of each of the considered vibrational levels of $N_2$ ($B^3\Pi_g$) around 10 ms after the sprite pulse[14].

On the other side, the identified metal traces result from the meteoric ablation of interplanetary dust particles entering the Earth's atmosphere at high speeds[17,41]. Neutral and ionic layers of meteoric

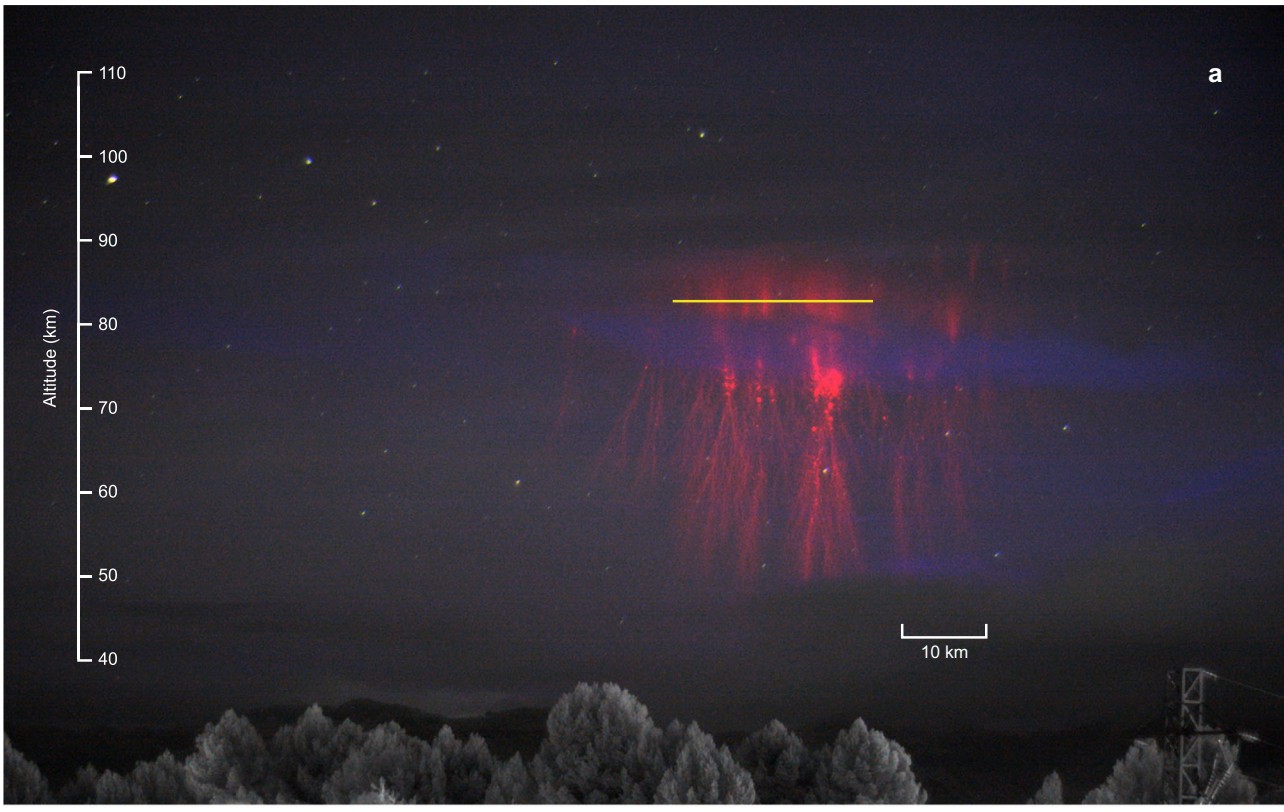

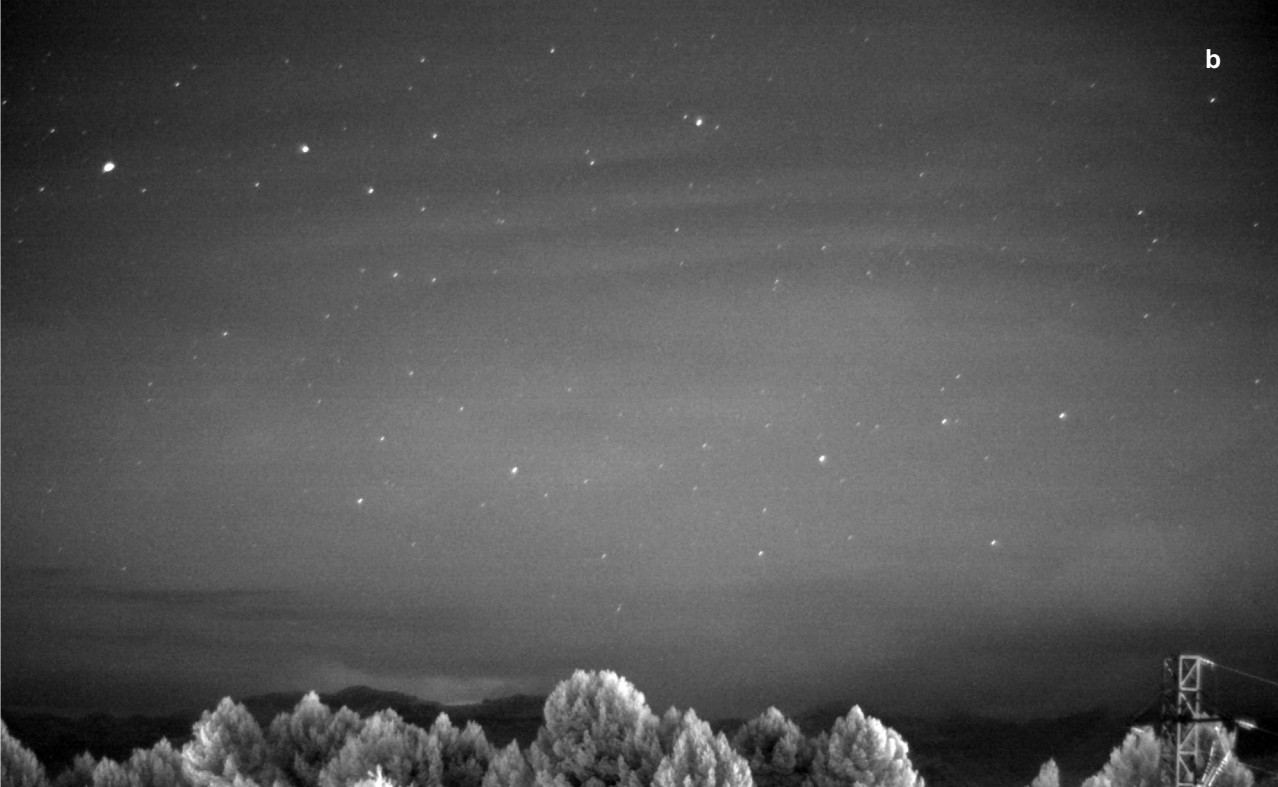

**Fig. 2 | Images recorded on 21 September 2019 at 19:45:14 UTC. a** Composite image of the jellyfish sprite (red). The slit projection is superimposed as a yellow line. Time-lapse video was taken with a monochrome Teledyne FLIR Grasshopper3 video camera (Sony IMX174 CMOS sensor) fitted with a 25 mm F0.95 lens and a 720 nm infrared long-pass filter. The exposure time of each frame was 1 s. The composite has been made using the sprite event frame as red RGB channel, the airglow image as the green channel, while the blue channel is a frame of a cirrus cloud revealed by the reflection of light from a distant lightning flash 13 s after the event. **b** Averaged (stacked) video frames of airglow background in the 10 s before and after the sprite time show some banded structure which we interpret as likely caused by modulation of the hydroxyl (OH') airglow layer by gravity waves.

**Table 2 | Set A. Identified species from −200 to 120 ms**

| Element | $\lambda$ (nm) | $A_{ki}$ (s⁻¹) | $\tau_{ki}$ (s) | Element | $\lambda$ (nm) | $A_{ki}$ (s⁻¹) | $\tau_{ki}$ (s) |
|---|---|---|---|---|---|---|---|
| Ni I | 503.54 | $5.70 \times 10^{7}$ | $1.75 \times 10^{-8}$ | Fe I | 567.18 | – | – |
| Fe I | 507.62 | – | – | Fe I* | 567.84 | – | – |
| Ni I | 515.80 | $5.90 \times 10^{5}$ | $1.69 \times 10^{-6}$ | Fe I* | 567.86 | – | – |
| Fe II | 520.60 | $2.1 \times 10^{5}$ | $4.76 \times 10^{-6}$ | Ni I | 568.22 | $7.80 \times 10^{6}$ | $1.28 \times 10^{-7}$ |
| Fe II* | 521.99 | $8.6 \times 10^{6}$ | $1.16 \times 10^{-7}$ | Fe I | 568.59 | – | – |
| Ni I* | 522.03 | $1.7 \times 10^{6}$ | $5.88 \times 10^{-7}$ | Fe I | 569.15 | $6.70 \times 10^{6}$ | $1.49 \times 10^{-7}$ |
| Fe I* | 522.06 | $5.7 \times 10^{-1}$ | 1.75 | Ni I | 569.50 | $1.70 \times 10^{7}$ | $5.88 \times 10^{-8}$ |
| O$_2^+$ | 523.90 | $4.00 \times 10^{3}$ | $2.50 \times 10^{-4}$ | Fe I | 569.61 | $4.35 \times 10^{5}$ | $2.30 \times 10^{-6}$ |
| Fe I | 525.19 | – | – | Fe I* | 569.84 | – | – |
| Fe I* | 525.57 | – | – | Fe II* | 569.86 | – | – |
| Ni I* | 525.64 | $2 \times 10^{6}$ | $5 \times 10^{-7}$ | Fe I* | 569.94 | – | – |
| Unidentified | 527.78 | – | – | Fe II* | 572.65 | $1.87 \times 10^{7}$ | $5.35 \times 10^{-8}$ |
| Fe I | 528.04 | $7.27 \times 10^{5}$ | $1.38 \times 10^{-6}$ | Fe I* | 573.98 | – | – |
| Fe I | 529.07 | $2.20 \times 10^{-1}$ | 4.55 | N I* | 573.99 | $7.49 \times 10^{5}$ | $1.34 \times 10^{-6}$ |
| N I | 531.03 | $9.40 \times 10^{4}$ | $1.06 \times 10^{-5}$ | N I* | 574.05 | $5.71 \times 10^{5}$ | $1.75 \times 10^{-6}$ |
| Fe II | 531.84 | $5.6 \times 10^{6}$ | $1.79 \times 10^{-7}$ | Na I | 574.45 | $4 \times 10^{6}$ | $2.5 \times 10^{-7}$ |
| O$_2^+$ | 533.60 | $2.4 \times 10^{4}$ | $4.17 \times 10^{-5}$ | Fe I | 575.40 | – | – |
| Fe I | 536.99 | $7.22 \times 10^{7}$ | $1.39 \times 10^{-8}$ | Fe I | 577.84 | $1.06 \times 10^{4}$ | $9.43 \times 10^{-5}$ |
| Fe I | 537.37 | $3.7 \times 10^{6}$ | $2.70 \times 10^{-7}$ | Fe I | 579.15 | – | – |
| Fe I | 538.75 | $3.03 \times 10^{5}$ | $3.30 \times 10^{-6}$ | Fe I | 579.39 | $6.10 \times 10^{5}$ | $1.64 \times 10^{-6}$ |
| Fe I | 544.13 | $5.00 \times 10^{5}$ | $2.00 \times 10^{-6}$ | Fe I | 583.46 | $9.00 \times 10^{-2}$ | $1.11 \times 10^{1}$ |
| Fe I | 545.21 | – | – | Fe I | 583.68 | – | – |
| Fe I | 545.54 | – | – | O$_2^{+\,*}$ | 584.10 | $8.48 \times 10^{+4}$ | $1.18 \times 10^{-5}$ |
| N I | 547.18 | $1.39 \times 10^{5}$ | $7.19 \times 10^{-6}$ | Fe I* | 584.11 | – | – |
| Fe I* | 547.66 | $8.70 \times 10^{6}$ | $1.15 \times 10^{-7}$ | Fe I | 584.32 | – | – |
| Ni I* | 547.69 | $9.50 \times 10^{6}$ | $1.05 \times 10^{-7}$ | Ni I* | 584.70 | $2.40 \times 10^{4}$ | $4.17 \times 10^{-5}$ |
| Fe I | 547.75 | $6.25 \times 10^{4}$ | $1.60 \times 10^{-5}$ | Fe I* | 584.81 | – | |
| Unidentified | 548.99 | – | – | Ni I* | 585.77 | – | |
| Fe I | 549.75 | $6.25 \times 10^{4}$ | $1.60 \times 10^{-5}$ | Fe I* | 586.72 | $2.10 \times 10^{-2}$ | $4.76 \times 10^{1}$ |
| N I | 553.00 | $1.78 \times 10^{5}$ | $5.62 \times 10^{-6}$ | Si II* | 586.75 | – | – |
| Fe I | 553.20 | $6.10 \times 10^{5}$ | $1.64 \times 10^{-6}$ | Fe I | 588.67 | – | – |
| Fe I* | 554.30 | – | – | Fe I | 588.75 | – | – |
| Fe I* | 554.32 | – | – | Na I | 589.00 | $6.16 \times 10^{7}$ | $1.62 \times 10^{-8}$ |
| Fe I | 554.70 | $9.60 \times 10^{5}$ | $1.04 \times 10^{-6}$ | O$_2^+$ | 590.00 | $1.17 \times 10^{5}$ | $8.55 \times 10^{-6}$ |
| Fe I | 556.12 | – | – | Fe I* | 591.41 | – | |
| Fe I | 556.27 | – | – | Fe I* | 591.42 | – | |
| Si II | 556.83 | $2.9 \times 10^{6}$ | $3.45 \times 10^{-7}$ | Fe I | 592.05 | – | – |
| Fe II | 557.15 | – | – | Fe I* | 593.38 | – | |
| O$_2^+$ | 557.50 | $1.54 \times 10^{4}$ | $6.49 \times 10^{-5}$ | Fe I | 595.27 | $1.50 \times 10^{6}$ | $6.67 \times 10^{-7}$ |
| O I | 557.73 | 1.26 | $7.94 \times 10^{-1}$ | Fe I | 596.22 | – | – |
| Fe I | 558.67 | $2.19 \times 10^{7}$ | $4.57 \times 10^{-8}$ | Fe I | 596.69 | – | – |
| Fe I* | 560.77 | – | – | Fe I | 597.53 | – | – |
| N$_2^*$ | 560.80 | $2.00 \times 10^{-5}$ | $5.00 \times 10^{4}$ | Fe I | 597.88 | – | – |
| Fe I* | 560.90 | – | – | Fe I | 598.23 | – | – |
| O$_2^{+\,*}$ | 560.90 | $3.52 \times 10^{5}$ | $2.84 \times 10^{-6}$ | Fe I | 599.37 | – | – |
| Fe I | 564.66 | – | – | Fe I* | 599.89 | – | – |
| Unidentified | 565.74 | – | – | | | | |

Detected species with at least two consecutive emission line brightness over the detection threshold (3σ) from −200 ms to 120 ms, and at least one emission line brightness higher than 5σ. $\lambda_i$ columns show the wavelengths (in nm). $A_{ki}$ columns show the atomic[25] and molecular[26] Einstein coefficients (in s⁻¹). $\tau_{ki}$ columns show radiative lifetimes (in s) of the related emission lines. Asterisks show blended lines. Middle dashes show unknown coefficients.

metals with thicknesses of the order of 10 km and concentrations between about $10^2$ and $10^5$ cm⁻³ are persistent near the mesopause, where a balance exists between the meteoric input, oxidation by atmospheric molecules, reduction of the latter by photolysis and by reaction with oxygen and hydrogen atoms and charge transfer processes. The most abundant meteoric metal in the upper mesosphere-lower thermosphere region is iron[41]. The iron layer peaks at 85 km on average, with a large density gradient at its bottom side which results from the proximity of the so-called chemical shelf[42]. The nighttime lower boundary of the iron layer is located between 80 and

**Table 3 | Set B. Identified species from 120 to 480 ms**

| Element | λ (nm) | $A_{ki}$ (s⁻¹) | $τ_{ki}$ (s) | Element | λ (nm) | $A_{ki}$ (s⁻¹) | $τ_{ki}$ (s) |
|---|---|---|---|---|---|---|---|
| Fe I | 507.63 | – | – | Fe I | 569.61 | $4.35 \times 10^5$ | $2.30 \times 10^{-6}$ |
| Fe I | 516.23 | – | – | Fe I* | 569.77 | – | – |
| Fe I* | 520.23 | – | – | Fe I* | 569.84 | – | – |
| Si II* | 520.24 | – | – | Fe II | 572.66 | $1.87 \times 10^7$ | $5.35 \times 10^{-8}$ |
| Fe I | 525.57 | – | – | N I | 574.05 | $5.71 \times 10^5$ | $1.75 \times 10^{-6}$ |
| O I | 527.52 | $4.89 \times 10^3$ | $2.04 \times 10^{-4}$ | Fe I | 574.75 | – | – |
| Fe I | 528.04 | $7.27 \times 10^5$ | $1.38 \times 10^{-6}$ | Fe I | 580.45 | $8.80 \times 10^{-2}$ | $1.14 \times 10^1$ |
| Fe I | 531.87 | – | – | Fe I | 580.99 | – | – |
| Fe I | 532.11 | $2.13 \times 10^6$ | $4.69 \times 10^{-7}$ | Fe I | 583.46 | $9.00 \times 10^{-2}$ | $1.11 \times 10^1$ |
| Fe I | 535.63 | $1 \times 10^{-3}$ | $1 \times 10^3$ | Fe I | 583.81 | – | – |
| Fe I | 536.75 | $7.13 \times 10^7$ | $1.40 \times 10^{-8}$ | Fe I | 584.11 | – | – |
| N I | 537.83 | $1.66 \times 10^5$ | $6.02 \times 10^{-6}$ | Ni I* | 584.7 | $2.40 \times 10^4$ | $4.17 \times 10^{-5}$ |
| Fe I | 538.22 | $7.9 \times 10^{-2}$ | $1.27 \times 10^1$ | Fe I | 585.46 | – | – |
| Fe I | 538.56 | $3.1 \times 10^4$ | $3.23 \times 10^{-5}$ | Fe I | 588.18 | – | – |
| Fe I | 540.21 | $5.6 \times 10^7$ | $1.78 \times 10^{-8}$ | Fe I | 588.75 | – | – |
| Fe I | 540.91 | $1.1 \times 10^6$ | $9.09 \times 10^{-7}$ | Fe I | 589.11 | – | – |
| O I | 543.52 | $7.74 \times 10^5$ | $1.29 \times 10^{-6}$ | $O_2^+$ | 590.00 | $1.17 \times 10^5$ | $8.55 \times 10^{-6}$ |
| Fe I | 545.21 | – | – | Fe I | 590.24 | – | – |
| Fe I | 547.85 | $6.8 \times 10^5$ | $1.47 \times 10^{-6}$ | Fe I | 592.05 | – | – |
| Ni I | 549.94 | $6.2 \times 10^4$ | $1.61 \times 10^{-5}$ | Fe I | 594.54 | – | – |
| N I | 557.58 | $3.42 \times 10^5$ | $2.92 \times 10^{-6}$ | Fe I | 594.75 | – | – |
| Ca I | 559.01 | $8.3 \times 10^6$ | $1.20 \times 10^{-7}$ | Fe I | 594.93 | – | – |
| Fe I* | 560.77 | - | - | Fe I | 595.82 | – | – |
| $N_2^*$ | 560.80 | $2.00 \times 10^{-5}$ | $5.00 \times 10^4$ | Fe I | 595.99 | – | – |
| Fe I* | 560.90 | - | - | Fe I* | 596.55 | – | – |
| $O_2^+$ | 560.90 | $3.52 \times 10^5$ | $2.84 \times 10^{-6}$ | Fe I* | 596.57 | – | – |
| Fe I | 562.40 | $8.3 \times 10^5$ | $1.20 \times 10^{-6}$ | Fe I | 597.53 | – | – |
| Si II* | 563.95 | – | – | Fe I* | 597.83 | – | – |
| Fe I* | 563.96 | $1.4 \times 10^{-1}$ | 7.14 | Fe I* | 597.89 | – | – |
| Fe I | 564.44 | – | – | Fe I | 598.85 | – | – |
| Fe I | 566.10 | – | – | $O_2^{+*}$ | 599.10 | $1.20 \times 10^3$ | $8.3 \times 10^{-4}$ |
| Fe I | 567.18 | – | – | Fe I* | 599.12 | – | – |
| Fe II | 567.53 | $1.37 \times 10^7$ | $7.30 \times 10^{-8}$ | O I | 599.53 | – | – |
| Si II* | 568.14 | $1 \times 10^7$ | $1 \times 10^{-7}$ | Fe I | 599.89 | – | – |
| Fe I* | 568.15 | – | – | – | | | |

Detected species with at least two consecutive emission line brightness over the detection threshold (3σ) from 120 to 480 ms, and at least one emission line brightness higher than 5σ. $λ_i$ columns show the wavelengths (in nm). $A_{ki}$ columns show the atomic[25] and molecular[26] Einstein coefficients (in s⁻¹). $τ_{ki}$ columns show radiative lifetimes (in s) of the related emission lines. Asterisks show blended lines. Middle dashes show unknown coefficients.

87 km and its hourly and daily variability is dominated by the linear response of metal density to wave perturbations, including tides and gravity waves[42]. The high variability of the bottom side of the meteoric metal neutral layers (and of the atomic oxygen shelf) might be related to the fact that not all energetic sprites develop a mesospheric ghost. Besides, the meteoric metal layers present a marked seasonal and latitudinal variability which might play a role in the occurrence of this phenomenon. Moreover, these metal traces may affect the triggering of transient luminous events[43,44], considering that the electron impact ionisation of metallic species could provide an abundance of seed electrons for sprite streamers to initiate[2,45,46].

We detected an overall increase of the spectral brightness of the ghost starting approximately 200 ms before the occurrence of the sprite, when the spectral brightness peaks, to abruptly decay until 120 ms afterwards, to slightly increase again until reaching its second peak at 320 ms and decreasing until its stabilisation at 480 ms. This trend is also observed in imaging analysis of mesospheric ghosts[8].

In the reported event, the peak currents of the lightning within the area of interest and a temporal window of 400 ms before and after the sprite parent stroke (178.2 kA), were too weak to generate sprites themselves in this case (Table 1). Nevertheless, these positive strokes might generate a halo[47,48], contributing to green emissions[2] before and after the sprite occurrence. This is due to the low concentration of species at that altitude, making the induced reduced electric field (E/N) non-negligible. According to the halo model by Parra-Rojas et al. (2013), lightning strokes with relatively modest peak currents (10–20 kA) can provide sufficient excited species (like O I($^1S_O$ - $^1D_2$)) able to produce detectable halo optical emissions[47].

On the other hand, the radiative lifetimes of the $N_2$ (VK) molecular and the O I and Fe I forbidden atomic emission lines are longer than the observed decay times, because these metastable species also undergo deactivating collisions with atmospheric molecules, especially below the quenching height (95 km for O I($^1S_O$ - $^1D_2$)), where quenching processes are severe[10,p. 119]. Moreover, it could also be reasonable to infer that after 480 ms the ghost emission is so weak that only very few photons reach the sensor through the slit and, therefore, the signal to noise ratio is not high enough to follow the complete decay.

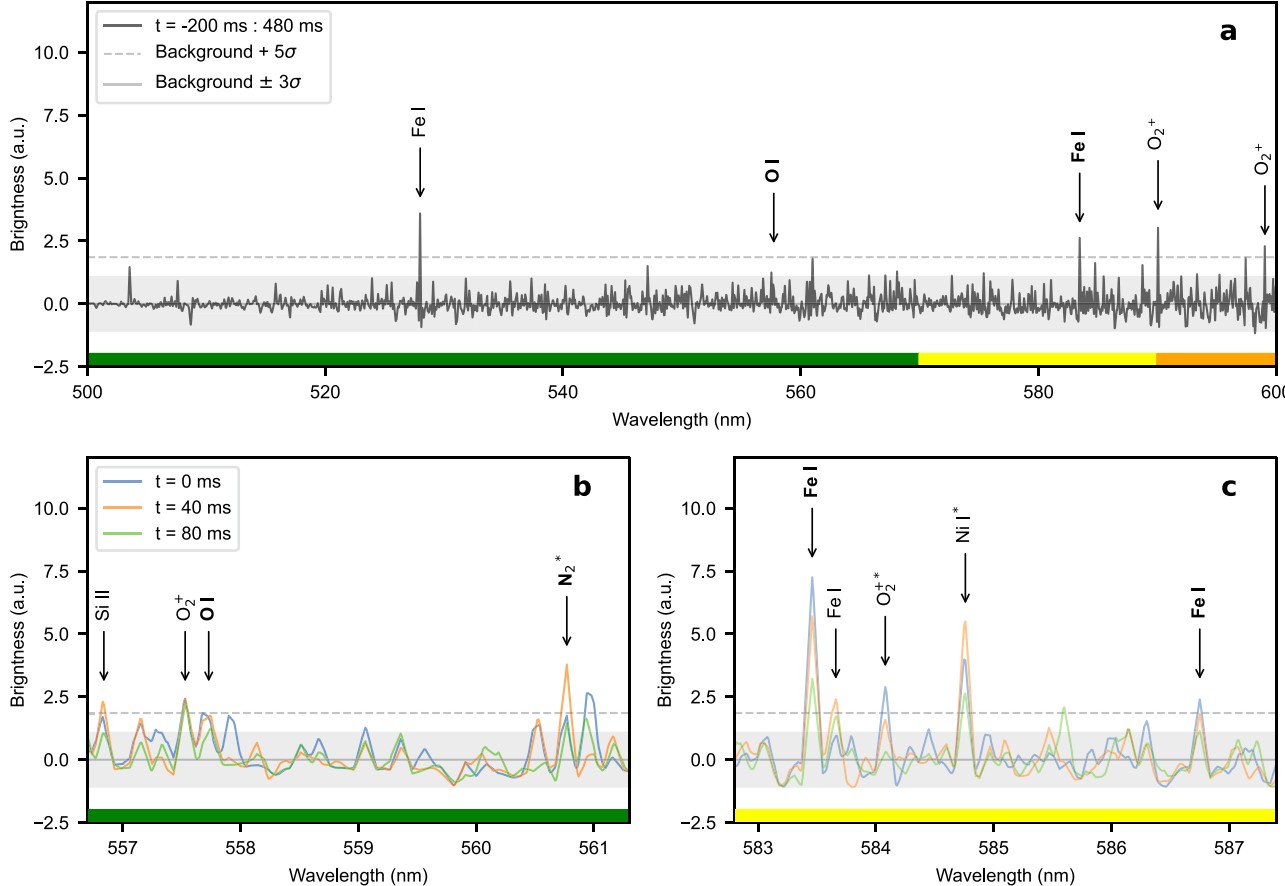

**Fig. 3 | Mesospheric ghost spectrum. a** Average of the consecutive reduced spectra of the reported event from 200 ms before the sprite to 480 ms after the sprite. **b**, **c** Zoom of the emission lines of interest from 0 ms (coincident with the sprite) to 80 ms after the sprite. Grey dashed lines show the $5\sigma$ confidence threshold; grey shadowed areas show the background level $\pm 3\sigma$ detection threshold. Thick solid horizontal bars in green, yellow and orange show the colour eye perception of the related wavelengths. We highlighted the identified forbidden emission lines with bold case labels. Asterisks show blended lines. Source data are provided as a Source Data file.

At this point one might think that, since the atomic iron emission lines at 580.45 nm, 583.46 nm and 586.72 nm belong to the yellow spectral range, the excited oxygen at 557.73 nm, the $N_2$ VK ($v' = 0$, $v'' = 15$) optical transition at 560.80 nm and the strongest emission lines found below 570 nm should be the only responsible for the slow decay of the green emissions of mesospheric ghosts. But the spectral response of the green component of most camera sensors usually covers a spectral range between 480 and 600 nm[49], including not only the green spectral range, but also yellow and orange. Therefore, the excited neutral iron at 580.45 nm, 583.46 nm and 586.72 nm can be detected by the green component of most camera sensors, so iron might be a strong candidate to contribute to the long lifetime greenish detection in ghost imaging, in combination with the well-known forbidden atomic oxygen and the $N_2$ VK metastable emissions at 557.73 nm and 560.80 nm, respectively.

Hence, since metal species, $O_2^+$ $(A^2\Pi_u - X^2\Pi_g)$, $O_2^+$ $(b^4\Sigma_g^- - a^4\Pi_u)$ and $N_2$ VK $(A^3\Sigma_u^+ - X^1\Sigma_g^+)$ are also responsible for the greenish emission of at least the event subject of this article, and not only the atomic oxygen emission line at 557.73 nm, we propose to rename the ghost acronym as GreenisH Optical emission from Sprite Tops.

The finding of metal traces in this ghost spectrum calls for an upgrade of current air plasma kinetic models, since to date, only species derived from hydrogen, oxygen, nitrogen, carbon and argon are taken into account to predict the spectral features of TLEs[12–15]. Moreover, further measurements combining imaging and spectroscopy are needed to build a reliable statistical basis of emitting species and boundary conditions inherent to the development of ghosts, to implement more accurate air plasma kinetic and electrodynamic models under the influence of TLEs.

## Methods

### Instrumentation

GRASSP[16–19] is an intensified slit spectrograph that records 2D spatial-spectral images in a wavelength range between 500 and 600 nm with a spectral resolution higher than 0.31 nm and 40 ms temporal resolution. A Watec monochrome camera 1/2" WAT-902H2 Ultimate triggers the spectrograph and helps to discern the origin of the spectrum. The spectrograph camera exhibits a 80 ms delay in its timestamp compared to the timestamp of the field camera. We take into account this delay to match the sequences of field and spectral images.

To record the images in Fig. 2, we used an additional high resolution video camera intended for other purposes: a PointGrey (Teledyne Flir) Grasshopper 3 USB3 GS3-U3-23S6M-C video camera with a monochrome Sony IMX174 sensor, a #F0.95 and 25 mm focal length lens and a long-pass filter at 720 nm. The composite image in Fig. 2 has been made using the sprite event frame as red RGB (Red-Green-Blue) channel, the airglow image as the green channel, while the blue channel is a frame of a cirrus cloud revealed by the reflection of light from a distant lightning flash 13 seconds after the event.

### Line identification

In this study, we employed custom Python scripts to reduce 45 raw 2D spatial-spectral consecutive images, from −480 ms to 1280 ms, being t = 0 ms the time coincident with the sprite. We followed standard data

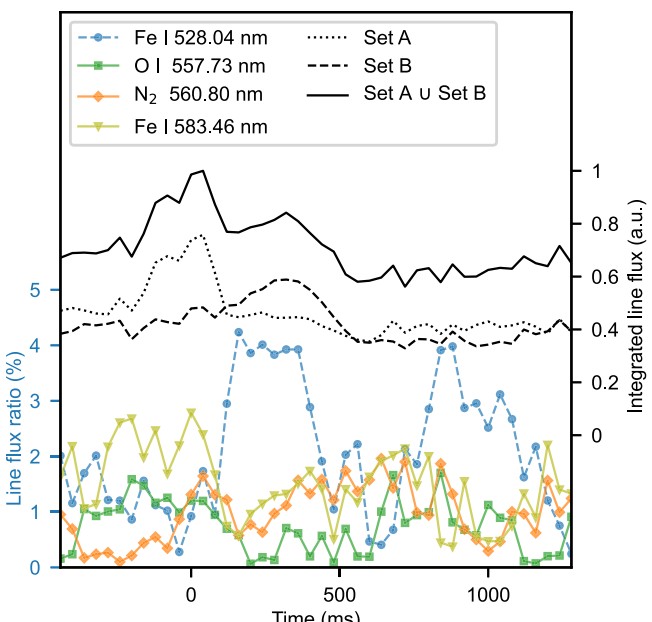

**Fig. 4 | Line flux evolution. Right axis.** Temporal evolution of the normalised average of the line fluxes of the identified species in Table 2 (set A, black dotted line), in Table 3 (set B, black dashed line) and in Tables 2 and 3 (combined set A and set B, black solid line). Left axis: Temporal evolution of the Fe I (528.04 nm), O I (557.73 nm), $N_2$ (560.80 nm) and Fe I (583.46 nm) line flux ratios. Note that t = 0 ms is coincident with the sprite. Source data are provided as a Source Data file.

reduction steps: bias subtraction, flat field correction, background subtraction, wavelength calibration and instrumental function correction. Then, for every frame, we integrated consecutive rows with brightness level higher than $3\sigma$ in the spatial dimension to enhance the signal-to-noise ratio of the spectral dimension, where $\sigma$ is the standard deviation of the background images.

Next, we superimposed all spectra to select those emission lines showing a brightness level higher than $5\sigma$ and higher than $3\sigma$ in at least two consecutive frames. We detected a global enhancement of the spectral brightness between -200 ms and 480 ms. We identified the emitting species in this temporal window by comparing the detected emission lines wavelengths to the well-known atomic[25] and molecular[26] spectral databases. We only considered those species with at least two identified emission lines, being one of them isolated.

## Line flux

We defined the observed flux of the emission line $n$ at wavelength $\lambda_n$ and instant $t$, as equation (1) shows:

$$F(\lambda_n, t) = \sum_{\lambda_i = \lambda_n - \Delta\lambda/2}^{\lambda_n + \Delta\lambda/2} S(\lambda_i, t) d\lambda, \tag{1}$$

where $S(\lambda_i, t)$ is the spectrum brightness (in a. u.) at wavelength $\lambda_i$ and instant $t$, $\Delta\lambda$ is GRASSP spectral resolution (0.31 nm) and $d\lambda$ is GRASSP spectral dispersion (0.13 nm/pixel).

We defined the line flux ratio (in percentage) of each emission line with respect to the average of all the identified emission lines fluxes between -200 ms to 480 ms as equation (2) shows:

$$R(\lambda_n, t) = \frac{F(\lambda_n, t)}{\sum_{i=1}^{N} F(\lambda_i, t)} \times 100N, \tag{2}$$

where $F(\lambda_n, t)$ is the observed flux of the emission line $n$ at wavelength $\lambda_n$ and instant $t$, calculated as equation (1), and $i$ represents each

detected emission line within the temporal window from −200 ms to 480 ms. $N$ is the total number of the detected emission lines between −200 ms to 480 ms (see Tables 2 and 3).

## Reporting summary

Further information on research design is available in the Nature Portfolio Reporting Summary linked to this article.

## Data availability

The source data are provided with this paper and in Figshare repository under accession code [https://doi.org/10.6084/m9.figshare.24161289]. The datasets generated during and/or analysed during the current study are available from the corresponding author on request. Source data are provided with this paper.

## Code availability

For Fig. 1, Meteosat satellite data has been downloaded from the EUMETSAT Earth Observation Portal [https://data.eumetsat.int], and Spinning Enhanced Visible and InfraRed Imager (SEVIRI) 10.8 $\mu m$ channel radiance data was converted and plotted as cloud top brightness temperature using QGIS software (v3.32). For Fig. 2, Fiji and ImageJ software were used to stack and RGB-composite images from the movie. For Figs. 3 and 4 Python custom scripts were used to reduce the spectral data and to calculate the line flux evolution. This code is available in Figshare repository under accession code [https://doi.org/10.6084/m9.figshare.24161289].

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

## Acknowledgements

We acknowledge the European Organisation for the Exploitation of Meteorological Satellites (EUMETSAT) for providing the Meteosat Second Generation cloud temperature data. The authors acknowledge Nowcast GmbH for providing LINET lightning data. The authors acknowledge C. Morosanu for helping to aim GRASSP. M.P.V., F.J.G.V. and J.S. acknowledge the financial support from the Spanish Ministry of Science and Innovation, MCIN, under projects PID2019-109269RB-C43 and PID2022-136348NB-C31, and FEDER program (M.P.V., F.J.G.V. and J.S.). M.P.V., F.J.G.V., J.C.G.M., J.S., F.J.P.I., R.S.R. and M.G.C. acknowledge the financial support from the Spanish Ministry of Science and Innovation, MCIN, under the research grant CEX2021-001131-S funded by MCIN/AEI/10.13039/501100011033 (M.P.V., F.J.G.V., J.C.G.M., J.S., F.J.P.I., R.S.R. and M.G.C.). F.J.P.I. acknowledges the financial support of a fellowship (LCF/BQ/PI22/11910026) from "la Caixa" Foundation (ID 100010434) (F.J.P.I.). O.V.V. and J.M. acknowledge the financial support from the Spanish Ministry of Science and Innovation, MCIN, under

project PID2019-109269RB-C42 and from the research grant ESP2017-86263-C4-2-R (O.V.V. and J.M.).

## Author contributions
M.P.V. performed data reduction, analysed spectral data, participated in the observing and led the paper writing. O.V.V. planned and coordinated observations, acquired the data, plotted the lightning map, plotted the composite image of the jellyfish sprite and estimated the sprite dimensions. F.J.G.V. conceived the experiments. F.J.G.V. and J.S. designed the experiments. J.S. and M.P.V. upgraded and calibrated GRASSP. M.P.V., F.J.G.V, M.G.C. and J.C.G.M. discussed the results. M.P.V., F.J.G.V., J.C.G.M. and F.J.P.I contributed to the writing and revision of this paper. R.S.R. and M.G.C. provided comments on the manuscript. J.M. provided LINET data.

## Competing interests
The authors declare no competing interests.

## Additional information

**Peer review information** : *Nature Communications* thanks the anonymous reviewers for their contribution to the peer review of this work. A peer review file is available.

