## [Peer Review File · Nature Communications]

Spectroscopy of a mesospheric ghost reveals iron emissionsEditorial Note: Parts of this peer review file have been redacted as indicated to avoid any copy right infringement.

REVIEWER COMMENTS

Reviewer #1 (Remarks to the Author):

This paper describes a high resolution optical spectrum recorded with good time resolution above a sprite where "green ghost" emission occurred which could be observed with impressive signal-to-noise. The spectrograph was located at an observatory near Barcelona, and the sprite occurred above a thunderstorm about 300 km away over the Mediterranean. The time-resolved spectrum recorded between 500 and 600 nm showed a number of spectral lines which - for the first time - allow the origins of the so-called "green ghost" to be determined. This is an emission that is infrequently observed above sprites. The authors show that the most important emission comes from metastable states of atomic Fe (these transitions are around 583 nm, and so visually yellow rather than green). The source of Fe is meteoric ablation, and the underside of the Fe layer coincides with the altitude where green ghost emission appears. The authors suggest that the reason that the emission is infrequent is because of the natural variability of the underside of the Fe layer, which seems reasonable.

The discovery that Fe is the major contributor to the green ghost is an important result, and should be published in a high profile journal. However, there are several aspects of the paper that are unclear or confusing, and so a major revision of the paper is needed before it should be considered for publication.

The first problem is the timing of the spectrum. According to lines 53 - 55, the spectrograph is triggered by a monochrome camera after a delay of 80 ms. So how can the spectra in Figure 3 refer to 0 ms after the sprite? And how can Figure 4 refer to -400 ms? Indeed, how is the emission shown in Figure 4 for 1000 ms, when we are told earlier in the paper that the spectrum was recorded for 400 ms?

The second point is the temporal evolution of the total emission in Figure 4, summarised in lines 74-77. These emissions are present before the sprite, and for at least 1000 ms afterwards (in fact, they appear to have stabilized). So how is this related to the sprite, in particular what causes the emission before the sprite? The authors show that several of the main emitters are species in metastable excited states with very long lifetimes, and explain the initial decay after 0 ms being due to quenching by collisions with air molecules. So why does that process stop after 500 ms?

A third point is that a number of molecular transitions (from O₂⁺ and N₂) are identified from specific $v' \rightarrow v''$ transitions. In order to confirm the identification of these molecular ions, surely more

vibrational transitions should be present between 500 and 600 nm? I would think at least 2 different $\nu' - \nu''$ transitions are needed for a positive identification.

More minor issues:

line 64: how can gravity waves in the mesosphere cause banded structures in cirrus clouds, which occur in the upper troposphere? What is the point of mentioning cirrus clouds here?

line 25: what are the "neutral metals" you refer to here? Are they not the Fe and Ni mentioned on line 24?

line 40: "a ghost"

line 45: "and so did we" is kind of obvious, and should be omitted.

line 72: "according to the star..."

line 73: "prevented us ..."

line 160-163: you state that metallic species might provide seed electrons. Given the concentration of metal atoms between 80 and 85 km is probably around $2 \times 10^4 \text{ cm}^{-3}$, if 100% of these were ionized would that be sufficient to initiate sprite streamers? That should be discussed.

line 167: "should be the only lines responsible for ..."

Reviewer #2 (Remarks to the Author):

The paper reports important discovery of iron emissions as responsible for green afterglow observed at tops of some (rare) lightning induced transient luminous events at mesospheric altitudes in the

Earth's atmosphere referred to as sprites. This a truly fascinating discovery and Nature Communications is an appropriate journal to transmit this important new knowledge to the community. I recommend publication after the authors address the following issues.

Major:

1. The temporal evolution in Figure 3 panel A is impossible to see and follow. There is a separate Figure 4 that is designed to show the temporal evolution. Can Figure 3 panel A be simplified to show only time integrated spectrum? This will make presentation sharper. Figure 4 can take responsibility to illustrate the time evolution features then.

2. The main result: the intensity of 583.46 nm Fe line is high before the sprite onset, then it falls after sprite onset, then is slowly rises, then it falls. The levels of intensity are lower after the sprite event in comparison with levels before the sprite event. This is what I see in Figure 4. My understanding is that the authors goal here is to illustrate that 583.46 nm increases in association with sprite, i.e. it is not just a background emission that exists there. Do the data shown in Figure 4 indicate this?

3. I do not understand the three lines on top of Figure 4. Lines supposed to cover different intervals i.e. 0-440 ms, 0-80 ms. I see they all go from -400 to 1000 ms. The paper should be self-contained and not requiring readers to refer to supplementary information to understand the content. Readers are sent to supplementary information on line 118 of the main text when description of these lines in Figure 4 is introduced. The authors may consider introducing a methods section, that is common to Nature journals, to cover these basics.

4. Lines 64-65 state "We observed thin cirrus clouds that night, showing some banded structure likely caused by gravity waves in the mesosphere (right panel of figure 1)." The gravity wave fields do modulate airglow layers (hydroxyl and 557.7 nm oxygen) that reside in altitude range 80-100 km that overlaps with sprite altitudes. There is a separate set of papers that document relationships of these gravity wave induced airglow modulations and sprites. Please explain what relevance does the image in the right panel of figure 1 have to the gravity wave airglow effects at mesospheric altitudes. If there is no direct connection I would suggest to remove that image.

Minor:

1. References. The document should include only standard citations (refer to peer reviewed documentation). I would suggest to remove facebook and youtube references. Also all AGU

abstracts/unpublished work can be removed. References 28, 29 and 30 can be removed as they do not add anything new to the report of this discovery. The reference 23 is a theory that has not been supported by observational evidence yet. Can be removed too. I would suggest to add important previous work talking about role of meteors with relation to sprites: Suszcynsky doi:10.1029/1999JD900962; Symbalisty doi:10.1006/icar.2000.6517. Also, about green emissions: Jehl doi:10.1029/2012JA018144. About green ghosts: Lyons doi:10.1080/00431672.2022.2116249. The references have misprints, in reference 1 it should read Red sprites, in reference 2 Blue jets, also TLE should be capitals in reference 13, in reference 22: N, W. Please correct.

2. There is only one section in this paper. The section titled "1.Text" looks awkward to me. I would suggest to remove this or introduce a more meaningful title if Nature Communication rules require one.

3. Line 86-87: "14 of the last ..." does not read right, please revise.

4. Line 96: N2 VK 0-15 should be referred as band (not line).

5. Line 98-99: "... states". May be "... upper states leading to corresponding transitions..."

6. Please also mark oxygen 557.7 nm in panel A of Figure 3.

7. Line 137: may be better to say "... and then to abruptly..."

8. Line 151: need a reference to substantiate that iron is the most abundant metal.

Dear Reviewers,

Thank you for your invaluable time and dedicated effort in reviewing our manuscript. Your commitment to thorough and feedback has been instrumental in shaping the final version of our work.

Below you can find a point-by-point response to your comments, in blue.

We restructured the manuscript to comply with the journal's guidelines, so it is different in shape from the original document. We hope that the manuscript is clearer now. If there is anything else we can add/remove to make it clearer, please, let us know.

Thank you again for your time and expertise.

Best regards,

María Passas Varo

REVIEWER COMMENTS

Reviewer #1 (Remarks to the Author):

This paper describes a high resolution optical spectrum recorded with good time resolution above a sprite where "green ghost" emission occurred which could be observed with impressive signal-to-noise. The spectrograph was located at an observatory near Barcelona, and the sprite occurred above a thunderstorm about 300 km away over the Mediterranean. The time-resolved spectrum recorded between 500 and 600 nm showed a number of spectral lines which - for the first time - allow the origins of the so-called "green ghost" to be determined. This is an emission that is infrequently observed above sprites. The authors show that the most important emission comes from metastable states of atomic Fe (these transitions are around 583 nm, and so visually yellow rather than green). The source of Fe is meteoric ablation, and the underside of the Fe layer coincides with the altitude where green ghost emission appears. The authors suggest that the reason that the emission is infrequent is because of the natural variability of the underside of the Fe layer, which seems reasonable.

The discovery that Fe is the major contributor to the green ghost is an important result, and should be published in a high profile journal. However, there are several aspects of the paper that are unclear or confusing, and so a major revision of the paper is needed before it should be considered for publication.

The first problem is the timing of the spectrum. According to lines 53 - 55, the spectrograph is triggered by a monochrome camera after a delay of 80 ms.

Sorry, there is a misunderstanding in this statement. We proceed to explain it better.

The spectrograph camera is triggered synchronously with the monochrome camera, but there is a delay of 80 ms of the spectrograph camera timestamp. We have clarified this in the revised manuscript. This means that the timestamp of the spectrograph camera is delayed 80 ms compared to the timestamp of the monochrome camera. So, although the trigger is synchronous, the timestamp is delayed. We found this out because the brightest images of the spectrum were coincident with empty images of the field camera, two frames after the parent sprite.

We did an experiment to confirm this: we aimed GRASSP with an intermittent source and plotted the evolution of the sum of all pixels of the images recorded by both cameras, being coincident both timestamps: Waterc and Lheritier stand for the monochrome and spectrograph cameras, respectively.

There is a delay of 80 ms between cameras, so we have to correct the timestamp of the 2D spatial-spectral images. Once this correction is made, we assign $t = 0$ ms to the time coincident with the parent sprite.

Y-axis: Sum of the brightness of all pixels for every frame
X-axis: Time in seconds

So how can the spectra in Figure 3 refer to 0 ms after the sprite?

Time $t=0$ ms is coincident with the sprite. We have clarified this within the manuscript.

And how can Figure 4 refer to -400 ms? Indeed, how is the emission shown in Figure 4 for 1000 ms, when we are told earlier in the paper that the spectrum was recorded for 400 ms?

There is a buffer of images before and after the triggering, so we can analyse previous and following frames. Indeed, we analyse the spectra between -480 ms to 1000 ms. We have clarified this in the revised manuscript.

The second point is the temporal evolution of the total emission in Figure 4, summarised in lines 74-77. These emissions are present before the sprite, and for at least 1000 ms afterwards (in fact, they appear to have stabilized). So how is this related to the sprite, in particular what causes the emission before the sprite?

That's a very good question. If you have a look at the lightning activity that night (see below), there were four positive lightning strokes in the same thunderstorm within a time gap of about 400 ms before the sprite, and five after the sprite. Their intensities were between 10 and 30 times less than the sprite parent stroke's. So, although these lightning strokes didn't develop a reduced electric field strong enough to trigger a sprite themselves, maybe the induced reduced electric fields were strong enough to trigger a halo. This can be explained because the reduced electric field (E/N) is not negligible, since the concentration of species is very low at that altitude. Indeed, according to the halo model by Parra-Rojas et al. (2013), "*Chemical and electrical impact of lightning on the Earth mesosphere: The case of sprite halos*", lightning strokes with relatively modest peak currents (10-20 kA) can provide sufficient excited species (like $O(1S)$) able to produce detectable halo optical emissions (see panels (d) and (e) of figure below). Also local electric fields might be generated by the metallic species and could contribute to the excitation process.

[redacted]

We have discussed this carefully within the manuscript.

LIGHTNING ACTIVITY BEFORE AND AFTER THE PARENT SPRITE

20190921 19:44:29,550	42,031	7,6545	0	1	-13,3	0,078
20190921 19:44:29,550	42,025	7,6465	0	1	-11,8	0,252
20190921 19:44:29,661	41,5831	6,9528	13,8	2	-8	0,123
20190921 19:44:29,665	41,5907	6,9457	0	1	6,5	0,226
20190921 19:44:29,765	42,0579	7,6244	0	1	4,4	0,268
20190921 19:44:29,765	42,0432	7,6044	0	1	-4,2	0,109
20190921 19:44:30,149	41,6559	7,0774	0	1	-4,8	0,296
20190921 19:44:30,150	41,6674	7,0486	0	1	10,5	0,074
20190921 19:44:30,372	41,5852	6,9506	0	1	15,1	0,147
20190921 19:44:30,403	41,7864	6,9443	0	1	13,4	0,194
20190921 19:44:30,403	41,7897	6,9567	0	1	13,5	0,222
20190921 19:44:30,514	41,9324	6,2828	0	1	178,2	0,106
20190921 19:44:30,548	41,8934	6,6062	0	1	7,1	0,072
20190921 19:44:30,550	41,88	6,615	0	1	18,5	0,132
20190921 19:44:30,608	41,9181	6,6422	0	1	4,7	0,284
20190921 19:44:30,629	41,8884	6,6076	0	1	5,5	0,061
20190921 19:44:30,629	41,8879	6,63	0	1	5,5	0,204
20190921 19:44:30,898	42,1914	6,6493	0	1	-4	0,395
20190921 19:44:31,029	41,9795	6,1373	0	1	-6,5	0,098
20190921 19:44:31,029	41,981	6,1718	0	1	-7,4	0,029
20190921 19:44:31,160	42,3882	7,6264	9,7	2	3,7	0,059
20190921 19:44:32,123	42,0152	7,5206	0	1	7,3	0,126
20190921 19:44:32,140	42,0077	7,514	10,1	2	4,1	0,067
20190921 19:44:32,263	41,985	7,4978	0	1	8,6	0,358
20190921 19:44:32,291	42,3938	7,6176	0	1	3,7	0,635

The authors show that several of the main emitters are species in metastable excited states with very long lifetimes, and explain the initial decay after 0 ms being due to quenching by collisions with air molecules. So why does that process stop after 500 ms?

Right, as you just mentioned, we stated that the radiative lifetimes of the identified species are longer than the observed decay times because these metastable species also undergo deactivating collisions with atmospheric molecules. In *Vallance Jones, A.: Aurora. D. Reidel Publishing Company, P.O.Box 17, Dordrecht, Holland (1974), pages 119-120*, it states that below the quenching height, the quenching processes are severe. This quenching height is 95 km for the O I (557.7 nm), being the dominant quenching particle O₂; and 145 km for the N₂ VK, being the dominant quenching particle O. Hence, as we are observing a phenomenon occurring at 83 km height, it is reasonable to agree that there are severe

quenching processes at this height, so the observed decay time is shorter than expected due to quenching.

Moreover, it could also be reasonable to infer that after 500 ms the ghost emission is so weak that only very few photons reach the sensor through the slit and, therefore, the signal to noise ratio is not high enough.

We added this discussion in the manuscript.

A third point is that a number of molecular transitions (from O₂⁺ and N₂) are identified from specific v' → v'' transitions. In order to confirm the identification of these molecular ions, surely more vibrational transitions should be present between 500 and 600 nm? I would think at least 2 different v' - v'' transitions are needed for a positive identification.

The criterion we used to identify spectral lines is to consider the species with at least two consecutive spectral emission line brightness over the detection threshold (3σ) from -200 ms to 480 ms, and at least one emission line brightness higher than 5σ in that time gap.

We found 4 molecular transitions related to O⁺ b⁴Σ⁻ - a⁴Π under this criterion: O

- + b⁴Σ⁻ - a⁴Π (1_v 0) 560.90 nm
- O₂⁺ b⁴Σ⁻ - a⁴Π (13, 11) 523.90 nm
- O₂⁺ b⁴Σ⁻ - a⁴Π_v(2, 2) 590.00 nmO
- + b⁴Σ⁻ - a⁴Π (9_v 10) 599.10 nm

Regarding to O₂⁺ A²Π_v - X Π_g besides the O₂⁺ A²Π_v - X Π_g(3, 15) 557.50 nm, and thanks to your review, we found that the O⁺ A²Π - X Π (2_g 15) 584.1 nm emission line is blended with Fe I 584.11 nm. We upgraded table 2 to add this blending.

Regarding N₂ VK, we only found one spectral line at 560.80 nm, blended with the Fe I (560.77 nm) spectral line. We didn't identify any other atom, ion or molecule surrounding this spectral line. We confirmed the N₂ VK (v' = 0, v'' = 15) (560.80 nm) emission line because previous modeling of the vibrational kinetics of air plasmas produced by the presence of

sprites in the mesosphere of the Earth at 78 km predicts the population of the vibrational level of N_2 ($A^3\Sigma^+_u$) to be dominant over the population of each of the considered vibrational levels of N_2 ($B^3\Pi_g$) around 10 ms after the sprite pulse, as you can read in Gordillo-Vázquez, F.J.: Vibrational kinetics of air plasmas induced by sprites. *Journal of Geophysical Research: Space Physics* 115(A5) (2010).

We found a weaker signature of N_2 VK (1,15) in 519.1 nm. Unfortunately, from all the sequence of spectra, only one emission line brightness was higher than 3σ (in $t = 80$ ms). That's the reason we didn't consider this spectral line (we didn't find any brightness of that spectral line higher than 5σ in the spectral evolution).

The black ellipse highlights the N_2 VK (1,15) in 519.1 nm emission line.

Besides Fe IV, we don't identify any other reasonable atom, ion or molecule emission line surrounding 519.1 nm. And Fe IV is unlikely to be observed. So we can infer that N_2 VK is a contributor for the greenish emissions.

More minor issues:

line 64: how can gravity waves in the mesosphere cause banded structures in cirrus clouds, which occur in the upper troposphere?

Thank you for this comment, there was an error in this sentence: "We observed thin cirrus clouds that night, showing some banded structure likely caused by gravity waves in the mesosphere (right panel of figure 1)."

We removed "in the mesosphere", because indeed, gravity waves due to thunderstorms start from the thunderstorm and propagate upwards beyond the mesosphere.

[redacted]

Liu, T.; Yu, Z.; Ding, Z.; Nie, W.; Xu, G. Observation of Ionospheric Gravity Waves Introduced by Thunderstorms in Low Latitudes China by GNSS. *Remote Sens.* **2021**, *13*, 4131. <https://doi.org/10.3390/rs13204131>

What is the point of mentioning cirrus clouds here?

As gravity waves pass through the cloud layer, they induce vertical motions, resulting in the stretching and compressing of the cloud particles. This can lead to changes in the cloud's vertical extent, horizontal organisation, and ice crystal alignment, altering the cloud's appearance and texture. The finding of this kind of structure within the cirrus clouds in the background images before and after the parent sprite suggests the presence of gravity waves that day, not only at tropospheric altitudes, but also propagating upwards beyond the mesosphere. If there is a presence of gravity waves, it is reasonable to understand the altitude variability of the mesospheric iron layer, which might affect to the green emission subject of the discussion in the manuscript. So the reason we added the right panel of figure 1 is to tell the reader that we are likely under a gravity wave scenario that might be related to ghosts.

line 25: what are the "neutral metals" you refer to here? Are they not the Fe and Ni mentioned on line 24?

We also found neutral Na as we show now in tables 2 and 3, and also ionic Si. We modified the abstract to be more accurate.

line 40: "a ghost" -> Ok

line 45: "and so did we" is kind of obvious, and should be omitted. -> Ok

line 72: "according to the star..." -> Ok

line 73: "prevented us ..." -> Ok

line 160-163: you state that metallic species might provide seed electrons. Given the concentration of metal atoms between 80 and 85 km is probably around $2 \times 10^4 \text{ cm}^{-3}$, if 100% of these were ionized would that be sufficient to initiate sprite streamers? That should be discussed.

Liu et al. "Formation of streamer discharges from an isolated ionization column at subbreakdown conditions" works in that direction. They estimate the requirements for the dimension and density of the small-scale isolated ionization patches for streamers initiation, by approximating the patch as a perfect cylindrical conductor. Under these conditions, they calculate that a cylindrical conductor of 3 m radius and between 21 and 52 m length with an electron density of around $3 \times 10^4 \text{ cm}^{-3}$ would initiate a streamer under a field on its tip of 3-5 times the conventional breakdown threshold.

Also, in Zabolin and Wright (2001) "Role of meteoric dust in sprite formation" they state that "*ubiquitous small conducting particles of meteoric origin in the mesosphere and stratosphere may explain some features of sprite occurrence and fine structure.*" They assume that, while in a vacuum the concentration of metals and the consequent amplified ambient field are insufficient for quick development of streamers to initiate (they refer to them as explosive emission), in the mesosphere there are air molecules of density $\approx 2 \times 10^{17} \text{ cm}^{-3}$ in the environment of these particles, and then the theory of arc discharges suggests several physical mechanisms that then may supply the required field amplification. However, this theory has not been supported by observational evidence yet. Indeed, we removed this reference from the manuscript according to another's referee comments.

line 167: "should be the only lines responsible for ..." -> Ok

Reviewer #2 (Remarks to the Author):

The paper reports important discovery of iron emissions as responsible for green afterglow observed at tops of some (rare) lightning induced transient luminous events at mesospheric altitudes in the Earth's atmosphere referred to as sprites. This a truly fascinating discovery and Nature Communications is an appropriate journal to transmit this important new knowledge to the community. I recommend publication after the authors address the following issues.

Major:

1. The temporal evolution in Figure 3 panel A is impossible to see and follow. There is a separate Figure 4 that is designed to show the temporal evolution. Can Figure 3 panel A be simplified to show only time integrated spectrum? This will make presentation sharper. Figure 4 can take responsibility to illustrate the time evolution features then.

You are right, we have modified this figure within the manuscript. Now we have plotted the integrated spectra between -200 ms and 480 ms. Thanks for helping for a better understanding of the article.

2. The main result: the intensity of 583.46 nm Fe line is high before the sprite onset, then it falls after sprite onset, then it slowly rises, then it falls. The levels of intensity are lower after the sprite event in comparison with levels before the sprite event. This is what I see in Figure 4. My understanding is that the authors goal here is to illustrate that 583.46 nm increases in association with sprite, i.e. it is not just a background emission that exists there. Do the data shown in Figure 4 indicate this?

That's a very good point. If you have a look at the lightning activity that night (see below), there were four positive lightning strokes in the same thunderstorm within a time gap of about 400 ms before the sprite, and five after the sprite. Their intensities were between 10 and 30 times less than the sprite parent stroke's. So, although these lightning strokes didn't develop a reduced electric field strong enough to trigger a sprite themselves, maybe the induced reduced electric fields were strong enough to trigger a halo. This can be explained because the reduced electric field (E/N) is not negligible, since the concentration of species is very low at that altitude. Indeed, according to the halo model by Parra-Rojas et al. (2013), "Chemical and electrical impact of lightning on the Earth mesosphere: The case of sprite halos", lightning strokes with relatively modest peak currents (10-20 kA) can provide sufficient excited species (like $O(1S)$) able to produce detectable halo optical emissions (see panels (d) and (e) of figure below). Also local electric fields might be generated by the metallic species and could contribute to the excitation process.

Figure 33. Altitude-time evolution of the $O(1S)$ density due to cloud-to-ground lightnings with (a) realistic current moment, (b) 100 kAkm peak current moment and 20 kAkm continuous current moment, (c) 200 kAkm peak current moment and 20 kAkm continuous current moment, (d) 100 kAkm peak current moment, and (e) 200 kAkm peak current moment.

We have discussed this carefully within the manuscript.

LIGHTNING ACTIVITY BEFORE AND AFTER THE PARENT SPRITE

20190921 19:44:29,550	42,031	7,6545	0	1	-13,3	0,078
20190921 19:44:29,550	42,025	7,6465	0	1	-11,8	0,252
20190921 19:44:29,661	41,5831	6,9528	13,8	2	-8	0,123
20190921 19:44:29,665	41,5907	6,9457	0	1	6,5	0,226
20190921 19:44:29,765	42,0579	7,6244	0	1	4,4	0,268
20190921 19:44:29,765	42,0432	7,6044	0	1	-4,2	0,109
20190921 19:44:30,149	41,6559	7,0774	0	1	-4,8	0,296
20190921 19:44:30,150	41,6674	7,0486	0	1	10,5	0,074
20190921 19:44:30,372	41,5852	6,9506	0	1	15,1	0,147
20190921 19:44:30,403	41,7864	6,9443	0	1	13,4	0,194
20190921 19:44:30,403	41,7897	6,9567	0	1	13,5	0,222
20190921 19:44:30,514	41,9324	6,2828	0	1	178,2	0,106
20190921 19:44:30,548	41,8934	6,6062	0	1	7,1	0,072
20190921 19:44:30,550	41,88	6,615	0	1	18,5	0,132
20190921 19:44:30,608	41,9181	6,6422	0	1	4,7	0,284
20190921 19:44:30,629	41,8884	6,6076	0	1	5,5	0,061
20190921 19:44:30,629	41,8879	6,63	0	1	5,5	0,204
20190921 19:44:30,898	42,1914	6,6493	0	1	-4	0,395
20190921 19:44:31,029	41,9795	6,1373	0	1	-6,5	0,098
20190921 19:44:31,029	41,981	6,1718	0	1	-7,4	0,029
20190921 19:44:31,160	42,3882	7,6264	9,7	2	3,7	0,059
20190921 19:44:32,123	42,0152	7,5206	0	1	7,3	0,126
20190921 19:44:32,140	42,0077	7,514	10,1	2	4,1	0,067
20190921 19:44:32,263	41,985	7,4978	0	1	8,6	0,358
20190921 19:44:32,291	42,3938	7,6176	0	1	3,7	0,635

3. I do not understand the three lines on top of Figure 4. Lines supposed to cover different intervals i.e. 0-440 ms, 0-80 ms. I see they all go from -400 to 1000 ms.

You are right, we have explained this better within the manuscript. Please let us know whether it is clearer now or not.

Let us highlight that we have enlarged the temporal axis, so we show the temporal evolution from -440 ms to 1280 ms.

The paper should be self-contained and not requiring readers to refer to supplementary information to understand the content. Readers are sent to supplementary information on line 118 of the main text when description of these lines in Figure 4 is introduced.

Ok, we have modified the manuscript in that direction.

The authors may consider introducing a methods section, that is common to Nature journals, to cover these basics.

Thanks for this comment. We re-organized the article to comply with the journal's guidelines. We hope it is clearer now.

4. Lines 64-65 state "We observed thin cirrus clouds that night, showing some banded structure likely caused by gravity waves in the mesosphere (right panel of figure 1)."

This sentence is confusing due to an error, we removed "in the mesosphere", because indeed, gravity waves due to thunderstorms start from the thunderstorm and propagate upwards.

Liu, T.; Yu, Z.; Ding, Z.; Nie, W.; Xu, G. Observation of Ionospheric Gravity Waves Introduced by Thunderstorms in Low Latitudes China by GNSS. *Remote Sens.* **2021**, *13*, 4131. <https://doi.org/10.3390/rs13204131>

The gravity wave fields do modulate airglow layers (hydroxyl and 557.7 nm oxygen) that reside in altitude range 80-100 km that overlaps with sprite altitudes. There is a separate set of papers that document relationships of these gravity wave induced airglow modulations and sprites.

Thanks, we added these references related to the modulation of the structure of cirrus clouds, sprites and elves due to gravity waves:

Haag, W., Kärcher, B.: The impact of aerosols and gravity waves on cirrus clouds at midlatitudes. *Journal of Geophysical Research: Atmospheres* 109(D12) D1202313 (2004)

Saha, S., Niranjan Kumar, K., Sharma, S., Kumar, P., Joshi, V.: Can quasi-periodic gravity waves influence the shape of ice crystals in cirrus clouds?. *Geophysical Research Letters* 47(11) e2020GL087909 (2020)

Pasko, V., Inan, U.S., Bell, T.F.: Sprites as evidence of vertical gravity wave structures above mesoscale thunderstorms. *Geophysical Research Letters* 24(14) 1735-1738 (1997)

Sentman, D.D., Wescott, E.M., Picard, R.H., Winick, J.R., Stenbaek-Nielsen, H.C., Dewan, E.M., Moudry, D.R., Sao Sabbas, F.T., Heavner, M.J., Morrill, J.: Simultaneous observations of mesospheric gravity waves and sprites generated by a midwestern thunderstorm. *Journal of Atmospheric and Solar-Terrestrial Physics* 65(5) 537–550 (2003)

Yue, J., Lyons, W.A.: Structured elves: Modulation by convectively generated gravity waves. *Geophysical Research Letters* 42(4) 1004–1011 (2015)

Please explain what relevance does the image in the right panel of figure 1 have to the gravity wave airglow effects at mesospheric altitudes. If there is no direct connection I would suggest to remove that image.

As gravity waves pass through the cloud layer, they induce vertical motions, resulting in the stretching and compressing of the cloud particles. This can lead to changes in the cloud's vertical extent, horizontal organisation, and ice crystal alignment, altering the cloud's appearance and texture. The finding of this kind of structure within the cirrus clouds in the background images before and after the parent sprite suggests the presence of gravity waves that day, not only at tropospheric altitudes, but also propagating beyond the mesosphere. If there is presence of gravity waves, it is reasonable to understand the altitude variability of the mesospheric iron layer, which might affect the green emission subject of the discussion in the manuscript. So the reason we added the right panel of figure 2 (previously figure 1) is to tell the reader that we are likely under a gravity wave scenario that might be related to ghosts.

Minor:

1. References. The document should include only standard citations (refer to peer reviewed documentation). I would suggest to remove facebook and youtube references. Also all AGU abstracts/unpublished work can be removed.

You are right, references to peer reviewed documentation should be the only taken into account to be cited in high impact journals. We removed Facebook references and all AGU abstracts/unpublished work.

On the other hand, as soon as this phenomenon is quite new, to date there is no peer reviewed documentation related to ghosts, excepting the article you mention below: "Inside the World of Sprite Chasing" (Walter Lyons, 2022). But as soon as we decided to analyse the spectral range between 500 and 600 nm after Hank Schyma and Paul Smith's YouTube video in 2019, we considered that this reference was necessary to draw a background to motivate the subject of our study. But if the editor considers that this information is not relevant, we will remove this reference.

References 28, 29 and 30 can be removed as they do not add anything new to the report of this discovery.

We cited these references to enhance that current TLEs chemistry models take into account only species derived from hydrogen, oxygen, nitrogen, carbon and argon to develop air plasma kinetic studies under the influence of TLEs. References 28-30 are three examples of such affirmation and under our opinion, they should be mentioned. But if the editor considers that this information is not relevant, we will remove these references.

The reference 23 is a theory that has not been supported by observational evidence yet. Can be removed too. -> Done

I would suggest to add important previous work talking about role of meteors with relation to sprites:

Suszczynsky doi:10.1029/1999JD900962; -> Ok

Symbalisky doi:10.1006/icar.2000.6517. -> Ok

Also, about green emissions: Jehl doi:10.1029/2012JA018144. -> Ok

About green ghosts: Lyons doi:10.1080/00431672.2022.2116249. -> Ok

The references have misprints, in reference 1 it should read Red sprites, in reference 2 Blue jets, also TLE should be capitals in reference 13, in reference 22: N, W. Please correct. -> Thanks very much. Done.

2. There is only one section in this paper. The section titled "1.Text" looks awkward to me. I would suggest to remove this or introduce a more meaningful title if Nature Communication rules require one. -> Done

3. Line 86-87: "14 of the last ..." does not read right, please revise. -> Done.

4. Line 96: N2 VK 0-15 should be referred as band (not line). -> Done.

5. Line 98-99: "... states". May be "... upper states leading to corresponding transitions..." -> Done

6. Please also mark oxygen 557.7 nm in panel A of Figure 3. -> Done

7. Line 137: may be better to say "... and then to abruptly..." -> Done

8. Line 151: need a reference to substantiate that iron is the most abundant metal. -> Done

REVIEWERS' COMMENTS

Reviewer #1 (Remarks to the Author):

Thank you for addressing the questions and comments that I posted in the initial review. The questions around timing and triggering of the spectrograph have been answered convincingly. The explanation for the Fe emission before the sprite being caused by a number of smaller lighting strokes is plausible. The Table showing an increased number of transitions between 500 and 600 nm makes the assignment of molecular ion transitions stronger. The more minor points have all been addressed satisfactorily. Note that the air density in the mesosphere is much lower than the value of $2 \times 10^{17} \text{ cm}^{-3}$ that you state. It varies from around 2×10^{16} to $2 \times 10^{14} \text{ cm}^{-3}$ between 48 and 80 km.

I recommend that the revised manuscript now be published.

Reviewer #2 (Remarks to the Author):

The paper can be accepted by the editor after the following minor corrections are made. I do not need to see the material again before it is published.

Line 75: suggest changing "likely" to "possibly" or "that might be". In other places, it is ok to say "likely" in Figure 2 caption. Also, "might" is ok on line 161.

Reason: Saying "likely" means you have some supporting evidence. There is no evidence provided: no altitude specified, no gravity wave frequency/period specified, no wavelength specified. The gravity wave has a Brunt-Vaisala frequency cutoff. Only waves with frequencies close to this Brunt-Vaisala frequency travel upward. For frequencies much less than this cutoff the waves propagate mostly horizontally. Also, it takes more than an hour to reach mesospheric altitudes and possibilities depend on vertical temperature profile significantly. The Brunt-Vaisala period is about 5 min. Additional experimental work, modeling and dispersion relationships are discussed in:

Taylor 1988 [https://doi.org/10.1016/0032-0633\(88\)90035-9](https://doi.org/10.1016/0032-0633(88)90035-9)

Dewan 1998 <https://doi.org/10.1029/98GL00640>

Snively 2013 <https://doi.org/10.1002/grl.50886>

I do not think this should hold the publication. It can be a good subject for additional research.

Line 110, and Figure 4 (two places): Instead of symbol "U" I would suggest to say "combined set A and set B"

In Table 2 I would suggest to replace "???" in three places with word "unidentified"

Recommend saying something to substantiate or simply removing on lines 211 and 212: "Also local electric fields might be generated by the metallic species and could contribute to the excitation process."

On Line 217 please include page numbers when books are cited i.e. book [10] in this case.

REVIEWERS' COMMENTS - 13/09/2023

Reviewer #1 (Remarks to the Author):

Thank you for addressing the questions and comments that I posted in the initial review. The questions around timing and triggering of the spectrograph have been answered convincingly. The explanation for the Fe emission before the sprite being caused by a number of smaller lighting strokes is plausible. The Table showing an increased number of transitions between 500 and 600 nm makes the assignment of molecular ion transitions stronger. The more minor points have all been addressed satisfactorily.

Note that the air density in the mesosphere is much lower than the value of $2 \times 10^{17} \text{ cm}^{-3}$ that you state. It varies from around 2×10^{16} to $2 \times 10^{14} \text{ cm}^{-3}$ between 48 and 80 km. -> True and corrected in the rebuttal file.

I recommend that the revised manuscript now be published.

Reviewer #2 (Remarks to the Author) 09/2023:

The paper can be accepted by the editor after the following minor corrections are made. I do not need to see the material again before it is published.

Line 75: suggest changing "likely" to "possibly" or "that might be". In other places, it is ok to say "likely" in Figure 2 caption. Also, "might" is ok on line 161.

Reason: Saying "likely" means you have some supporting evidence. There is no evidence provided: no altitude specified, no gravity wave frequency/period specified, no wavelength specified. -> Thanks for that point. You are right, we don't have any measurements to confirm 100% the presence of gravity waves. After your first comments, we isolated the sprite event, a lightning-illuminated cirrus cloud and the airglow background from the monochrome images and made a RGB composite to show their relative positioning. This is important in order to confirm that the spectrograph slit is observing the sprite, not any reflection from the cirrus cloud. The presence of the gravity waves may possibly promote the appearance of a green ghost on sprite tops by enhancing the Fe ion density in the altitude range of the sprite. The 720 nm filter of the field camera targets OH* airglow (at the cost of removing some sprite light). Find below two images of GWs that are caused by other thunderstorms to appreciate the similarity. The banded structure visible in the upper part of both panels of figure 2 shows a slight curvature consistent with concentric gravity waves propagating away from the convective core of the thunderstorm system that produced the sprites. This is the reason we consider that we were likely under a GW scenario. We have explained it carefully in the manuscript.

The gravity wave has a Brunt-Vaisala frequency cutoff. Only waves with frequencies close to this Brunt-Vaisala frequency travel upward. For frequencies much less than this cutoff the waves propagate mostly horizontally. Also, it takes more than an hour to reach mesospheric altitudes and possibilities depend on vertical temperature profile significantly. The Brunt-Vaisala period is about 5 min. Additional experimental work, modeling and dispersion relationships are discussed in:

Taylor 1988 [https://doi.org/10.1016/0032-0633\(88\)90035-9](https://doi.org/10.1016/0032-0633(88)90035-9)

Dewan 1998 <https://doi.org/10.1029/98GL00640>

Snively 2013 <https://doi.org/10.1002/grl.50886>

I do not think this should hold the publication. It can be a good subject for additional research.

Line 110, and Figure 4 (two places): Instead of symbol “U” I would suggest to say “combined set A and set B” -> We replaced the U symbol by “combined set A and B” within the text. However, we kept the U symbol in the legend of Figure 4, since the text “combined set A and B” is too long.

In Table 2 I would suggest to replace “???” in three places with word “unidentified” -> Done

Recommend saying something to substantiate or simply removing on lines 211 and 212: “Also focal electric fields might be generated by the metallic species and could contribute to the excitation process.” -> Done

On Line 217 please include page numbers when books are cited i.e. book [10] in this case. -> Done